# A Comparative Study of Water Quality and Human Health Risk Assessment in Longevity Area and Adjacent Non-Longevity Area

**DOI:** 10.3390/ijerph16193737

**Published:** 2019-10-04

**Authors:** Jiawen Yu, Jinlong Zhou, Aihua Long, Xinlin He, Xiaoya Deng, Yunfei Chen

**Affiliations:** 1College of Water and Architectural Engineering, Shihezi University, Shihezi 832000, China; yujiawen_415@163.com (J.Y.); hexinlin2002@163.com (X.H.); 2State Key Laboratory of Simulation and Regulation of Water Cycle in River Basin, Department of Water Resources, China Institute of Water Resources and Hydropower Research, Beijing 100038, China; lily80876@163.com; 3College of Water Conservancy and Civil Engineering, Xinjiang Agricultural University, Urumqi 830052, China; zjzhoujl@163.com (J.Z.); chenyunfi89@163.com (Y.C.); 4Xinjiang Hydrology and Water Resources Engineering Research Center, Urumqi 830052, China

**Keywords:** longevity area, water quality, groundwater quality, water environment, health risk, Xinjiang

## Abstract

A longevity area in Xinjiang, China and an adjacent non-longevity area both have similar climatic and hydrogeological conditions, and the residents of the two control groups have similar ethnic composition, diets and lifestyles. This study investigated if differences in groundwater quality between the longevity area and the non-longevity area are associated with the health of residents in the two control groups. In order to quantitatively describe the groundwater quality of the two control groups and its influence on human health, the Fuzzy Comprehensive Evaluation Method (FCEM) was used to compare and assess the overall water environment of the two control groups. Furthermore, the human health risk of groundwater for the two control groups was assessed using the Health Risk Assessment Model recommended by the U.S. Environmental Protection Agency (USEPA). Results showed that the overall water environment categories for the longevity area and non-longevity area are moderate quality (grade III) and very poor quality (grade V), respectively. The main health risk in the longevity area water environment is the non-carcinogenic risk (*HQ_LLV_*) caused by Cl^−^. The main health risks in the non-longevity area water environment are the non-carcinogenic risk (*HQ_CA_*) caused by Cl^−^ and the carcinogenic risk (*Risk_CA_*) caused by As. The total health risk (*HR_all_*) caused by over-standard inorganic pollutants in the water environment of the non-longevity area is 3.49 times higher than that of the longevity area. In addition, the study showed that the water environment pollution downstream of the Keriya River is conjunctively caused by agricultural activities and domestic sewage. The overall water environment of the longevity area is more conducive to the health-longevity of residents than the non-longevity area.

## 1. Introduction

With the rapid development of China’s social economy, living standards, and medical and health care services are improving continuously. China’s population structure has also undergone tremendous changes; the most prominent feature being population aging, which reached a peak in 2000 [1]. Therefore, the relationship between population aging and health has gained much attention from the government and the public in recent years [2,3]. For example, in May 2013, the China Health and Retirement Longitudinal Study (CHARLS) project of the National School of Development (NSD) at Peking University concluded that “the result of population aging depends largely on whether it can achieve healthy aging”. There are many healthy aging areas (longevity areas) in China [4,5]. It has important practical significance in studying the relationship between the environmental characteristics and population health-longevity in these areas for promoting the overall healthy aging of the country. Groundwater is the main source of water in many longevity areas of the world, and the water environment is the most basic, active and broad influential factor in sustaining human life [6]. Therefore, with research expanding on groundwater quality evaluation and risk assessment in recent years, the impact of the groundwater environment on human health-longevity has gained much attention from the public [7,8]. For example, scholars have used different groundwater environmental quality assessment and risk assessment methods to conduct considerable research on longevity areas in Pakistan [9], Caucasus [10], and, in China, Hainan county [11], and Xiayi county as well as the non-longevity area of Yucheng county [12]. These studies have provided abundant experiences and methods for water environment analysis and quality control.

In this paper, the Layisu Longevity Village (LLV) in Xinjiang, China and the adjacent control area (CA) were studied. LLV was recognized in November 1985 by the International Natural Medical Association as one of five villages with the highest longevity in the world [13]. LLV and CA are typical inland arid areas located southwest of Xinjiang with a dry climate and scarce water resources. Groundwater is the dominant source of domestic and agricultural water in the two control groups [6]. Therefore, groundwater quality and safety are expected to have an important impact on the health of local residents. This paper has applied the method of combining water quality assessment and health risk assessment to comprehensively evaluate the water environment that affects the health and longevity of residents in the research area. There are many intelligent evaluation methods for evaluating water environment [14,15,16]. In this study, the fuzzy comprehensive evaluation method was adopted to systematically evaluate the groundwater quality of the two control groups. The objectives of this paper are (1) to investigate and analyze the structural characteristics of the elderly population, and the diet and lifestyle of residents in the two control groups; (2) conduct a comparative analysis of different characteristics of inorganic components in their groundwater; (3) conduct a comparative assessment of the groundwater quality and human health risk; and (4) discuss the influence of these inorganic components on the residents’ health in order to provide basic information and a scientific basis for the residents’ longevity.

## 2. Study Area

LLV (E 81°14′48″~E 81°23′00″, N 36°49′55″~N 36°57′58″) is located southeast of Taklimakan Desert, north of Kashtash Mountain, and west of Yutian County Town, covering an area of 422.47 km^2^. LLV is located in the hinterland of Eurasia, with many springs flowing out of a exudation. The total spring flow is 1120 × 10^4^ m^3^/year.

The largest spring is the Layisu Spring (located on the north side of LLV), which has a flow of 1.02 × 10^4^ m^3^/year [17]. Its recharge source is the Holocene Series (Q4) pore unconfined water infiltration in the southern Kashtash Mountain area [2]. The CA (E 81°00′~E 82°30′, N 36°18′~N 37°18′) is a non-longevity area adjacent to LLV in Yutian County. The CA area is 3.90 × 10^4^ km^2^. The climatic factors, hydrogeological conditions and economic contrasts of the CA are similar to LLV (Table 1) [18,19].

## 3. Materials and Methods

### 3.1. Investigation and Sample Processing

In October 2017, a field survey of LLV was conducted to investigate the diet and lifestyle of local residents. In addition, the census data collected by questionnaire surveys and interviews, were statistically analyzed. In LLV, four groundwater samples (sampling numbers, L1–L4) were collected from different sampling sites in March 2014. L1: drinking water sampling point. L2–L4: the agricultural water sampling points. In CA, 15 groundwater samples (all unconfined groundwater samples, with numbers, H11–H25) were taken from different sampling sites in June 2014. H12 and H15–H17: the drinking water sampling points., H13, H14 and H18–H25: the agricultural water sampling points number (Figure 1). Moreover, four groups of samples were collected from each sampling site. Each group of samples included a bottle of 550 mL acidified water sample (5 mL concentrated nitric acid was added immediately after sampling as protective agent) and a bottle of 1500 mL original sample. The agricultural land and residential area in the study area are regularly distributed along the national road. The water intake points for agricultural production and living of residents are both come from the sampling points in our research. The distribution of the most sampling points along national road could be better reflected the direct effect of local groundwater environment on residents’ health.

Sample collection, preservation and detection were performed in strict accordance with China’s Specification for Regional Groundwater Contamination Survey and Evaluation (DZ/T0288-2015). Water samples were filtered with 0.22 μm disposable filter membranes to remove approximately 99.5% percent of colloids and particles. Water samples for major cation and trace element analysis were collected in 550 mL polyethylene bottles washed by HNO_3_ and acidified with 1:1 HNO_3_ to pH < 2. Samples for anion analysis were filtered without addition of reagents. The samples were preserved at 4 °C using an incubator and ice before being transported to the laboratory for analysis, and field blank samples and parallel samples were collected to assess sampling reliability. The levels of inorganic components in the samples were determined at the Mineral Water Testing Center of the Institution of Hydrogeology and Environmental Geology, Chinese Academy of Geological Sciences, which has been qualified by the China National Accreditation Service for Conformity Assessment. All sample analysis was carried out according to China’s National Food Safety Standard and Drinking Natural Mineral Water Test Method (GB/T8538-2016). The components measured included field test indexes (pH values) and inorganic indexes (K^+^, Na^+^, Ca^2+^, Mg^2+^, Cl^−^, I^−^, F^−^, SO_4_^2−^, HCO_3_^−^, NH_4_^+^, NO_3_^−^, NO_2_^−^, total dissolved solids [TDS], total hardness [TH], total Fe [TFe], Cu, Pb, Zn, Mn, Cr^6+^, Cd, Hg, As, Se, Br, Li, Sr, H_2_SiO_3_ and free CO_2_). Detailed measurement methods and detection limits are shown in Table 2. The reliability of the sampling data of this study was tested by the cations-anions balance method: when %CBE < ±5%, the physicochemical analysis is reliable. The %CBE of the cations-anions of water samples in this study area were −1.63~2.30%, indicating that all the data were reliable [20].

### 3.2. Water Hydrochemistry Types

Water hydrochemistry types were classified with the Shukalev classification method [21]. According to six major ion types (Na^+^, Ca^2+^, Mg^2+^, Cl^−^, HCO_3_^−^ and SO_4_^2−^; K^+^ was incorporated in Na^+^) and TDS in groundwater, anions and cations with concentrations over 25% mmol were combined to classify water types (a total of forty-nine types). Each water type represented the natural water formed in a specific environment [20]. In the Shukalev classification diagram, the direction from the upper left corner to the lower right corner generally indicates a transition from low mineralized water to high mineralized water [22,23].

### 3.3. Water Environment Quality Assessment

The Fuzzy Comprehensive Evaluation Method (FCEM) was applied for groundwater quality assessment in the present [24]. The FCEM comprehensively considers the fuzziness of water quality classification limits by constructing the membership function. Through the functional relationship, water environment factors with unclear classification limits that are difficult to quantify are quantified [25].

The Groundwater Quality Standard (GB/T14848-2017) classifies groundwater quality into five levels: grade I (excellent), grade II (good), grade III (moderate), grade IV (poor) and grade V (very poor). Grade I-III is suitable for all applications. Grade IV is poor quality and can affect human health adversely after direct drinking. Grade V water is unsafe to use for any purpose. This study assigns the inorganic chemical components with high detection rates over the standard as the main assessment factors. It establishes the corresponding assessment sets (*V_Standard limit_* = {*y*_1_,*y*_2_,*y*_3_,*y*_4_,*y*_5_}). By grading the limit value intervals of assessment factors, the grades were based on priority, and the highest grade among the individual inorganic components was used as the assessment result.

Most of the current water quality standards are based on the unidirectional distribution of concentration values. Therefore, this study adopts the “low-half-trapezium distribution function method” to determine the corresponding membership function [26]. The membership function is represented by *y_i_*(*x*). The maximum membership is 1.000 and the minimum is 0.000. The membership functions as follows:

Water quality belonging to grade1 is:(1)$$y_{1}(x)=\{\begin{array}{cc} 1 & x\leq x_{1}\\ \frac{x_{2}-x}{x_{2}-x_{1}} & x_{1}<x<x_{2}\\ 0 & x\geq x_{2} \end{array}$$

Water quality from grade *m* is (here, *m* is 2 to 4):(2)$$y_{m}(x)=\{\begin{array}{cc} 1 & x=x_{m}\\ \frac{x-x_{m-1}}{x_{m}-x_{m-1}} & x_{m-1}<x<x_{m}\\ \frac{x_{m+1}-x}{x_{m+1}-x_{m}} & x_{m}<x<x_{m+1}\\ 0 & x\leq x_{m-1}orx\geq x_{m+1} \end{array}$$

Water quality from grade 5 is:(3)$$y_{5}(x)=\{\begin{array}{cc} 1 & x\geq x_{5}\\ \frac{x-x_{4}}{x_{5}-x_{4}} & x_{4}<x<x_{5}\\ 0 & x\leq x_{4} \end{array}$$

*y_i_*(*x*) is the membership degree of each evaluation factor measured value between two adjacent intervals of water quality standards. $U_{Assessment\;factors}$ Cand the membership degree of *V_Standard limit_* = {*y*_1_,*y*_2_,*y*_3_,*y*_4_,*y*_5_} constitutes a (m × 5)-th order fuzzy matrix $R_{L1}$.

The weights were the contribution of each assessment factor to environmental pollution. The weight of each factor was calculated by the super weighting method. First, the pollution index of each factor was calculated. Second, the pollution index of each factor was normalized. Finally, the (1 × m)-th order weights matrix *B_i_* of the assessment factors was obtained.

The weights matrix *B_i_* of each factor and the membership fuzzy matrix *R_i_* were weighted and combined to obtain a (1 × 5)-th order comprehensive membership matrix *A_i_* of the water quality. According to the principle of maximum membership degree, the level of the maximum membership degree in the matrix *A_i_* was taken as the water quality category of the sampling point [27].

### 3.4. Human Health Risk Assessment

Human health risk assessments estimate the nature and probability of adverse health effects on humans from exposure to excessive pollutants [28]. This study used the U.S. Environmental Protection Agency Health Risk Assessment Model (USEPA-HRAM) to estimate the dosage of over-standard inorganic chemical non-carcinogens and carcinogens ingested when residents drink groundwater (Drinking water is the most direct way for recipients to ingest pollutants in water) [29].

The non-carcinogenic risk assessment model (*HQ_nj_*) and carcinogenic risk assessment model (*Risk_ci_*) generated by the drinking water exposure pathway are as follows:(4)$$HQ_{nj}=\frac{10^{-6}\times w\times C_{nj}}{A\times RfD_{nj}\times L}$$
(5)$$Risk_{ci}=\frac{1-\exp(-\frac{w\times C_{ci}\times q_{ci}}{A})}{L}$$
where *HQ_nj_* (hazard quotient) and *Risk_ci_* are the annual average non-carcinogenic risk and carcinogenic risk value of a single pollutant, respectively [year^−1^]; *w* is daily drinking water intake for adults [L/day]; *C_nj_*/*C_ci_* is the non-carcinogenic/carcinogenic pollutant concentration in drinking water [mg/L]; *A* is adult per capita weight in the study area [kg]; *L* is life expectancy of population [year]; *RfD_nj_* is the reference dosage for non-carcinogenic risk through the drinking water exposure pathway [mg/(kg·day)]; *q_ci_* is the cancer slope factor for carcinogenic risk [(kg·day)/mg] [30]. In this study area, according to the *Exposure Factors Handbook of Chinese Population* (Ministry of Environmental Protection of the PRC 2013), the daily drinking water intake for adults (w) is 2.8 L/day, with an adult per capita weight (A) of 62.1 kg. Assuming that there is no antagonistic or synergistic effect of each pollutant toxicity, the total health risk value (*HR_all_*) produced by the pollutants in the groundwater through the drinking water exposure pathway is the sum of *HQ_n_* and *Risk_c_*. The equations are as follows:(6)$$HQ_{n}=\sum_{j=1}^{k}HQ_{nj}\;\;Risk_{c}=\sum_{i=1}^{m}Risk_{ci}$$
(7)$$HR_{all}=HQ_{n}+Risk_{c}$$

The non-carcinogenic properties and carcinogenic properties of pollutants were determined according to the reliability classification of the International Agency for Research on Cancer (IARC) and World Health Organization (WHO) on the carcinogenicity of various inorganic pollutants in the water environment [31,32]. *RfD_nj_* and *q_ci_* reference values are from the U.S. EPA Guidelines for carcinogen risk assessment [32].

## 4. Results and Discussion

### 4.1. Population Structural Characteristics and Investigation

Table 3 shows the comparative statistical analysis of population status between LLV and CA (Statistics at the end of 2017). The study area is predominantly inhabited by the Uygur ethnic group. The longevity level index represents the ratio between the population above 80 years to those above 60. The 100-year-old level index is the ratio between the population above 100 years to those above 90 [33]. In LLV, the population aged 60 and above accounts for 8.3% of the total population; the longevity level index was 10.0%; the 100-year-old level index was 25.0% (maximum age: 103 years). In CA, the population aged 60 and above accounts for 6.9% of the total population; the longevity level index was 7.6%; the 100-year-old level index was 8.9% (including 14 people over 100 years old). The population aged 100 and above in LLV accounts for 1.03/10^3^ of the total population, which was 14.73 times higher than the accreditation standards of a “Longevity Township” set by the China Association of Gerontology and Geriatrics, and 21.06 times higher than CA. In addition, the weighted mean age of the population of all ages in LLV was 1.98 years elder than CA. The child mortality rate in LLV was also 8.96‰ lower than that in the CA. In terms of spatial distribution, the total population size, population over 60 year old and over 80 year old were only accounted for 1.1%, 1.0% in the LLV, and those of the proportion of the people 60 year old and over 80 year old were over 1.2% and 1.7% in the CA. While, the number of centenarians in LLV was 21.4% of the CA. Data indicate that the population aggregation of centenarians in LLV was much higher than that in the CA. This difference was substantial. According to the relevant data from Yutian County bureau of statistics, the population average life expectancy, mean population age and other “absolute indicators” of the two adjacent control groups were higher than the national average. However, the relative population aggregation of centenarians in LLV was much higher than that in CA. Therefore, the longevity area with the non-longevity area was the concept of a “relative”. This can be seen to result from the comparative analysis of the research results by Wang et al., [34] and Xiong et al., [35].

Differences in diet and lifestyle can effect residents’ health and longevity [37]. The diet and lifestyle of residents in LLV and CA were investigated and determined to be basically similar. Staple foods were mainly nang (scones made from wheat flour), ramen, huajuan (pasta steamed with wheat flour), lamb pilaf and cornmeal; vegetables were mainly onions, cabbage, potatoes and carrots; fruits and nuts were mainly apricots, apples, grapes, peaches, red dates and walnuts (all of which are locally produced); and meat-poultry were mainly mutton, eggs, milk, beef and chicken. In the two control groups, the elderly avoided cold and spicy foods, but Uygur medicinal teas (mainly consisting of brick tea with *Cistanche deserticola*, mint, clove, celery seed, valerian, cardamom, galangal, pepper and licorice) were consumed after meals. Long-term consumption of medicinal tea has pharmacological effects for anti-fatigue, anti-reserpine-induced stomach ulcers, and maintenance of liver functions and dieresis [38]. In addition, flavonoids (vitexin and morin) in medicinal tea inhibit the proliferation of human tumor cells [39]. The elderly of LLV and CA generally consumed medicinal tea at noon daily. Because the diet and lifestyle of the two control groups were basically similar, and both groups primarily used groundwater for drinking water and agricultural water, the difference in the water quality may be one of the factors associated with longer life spans and fewer health concerns of residents in LLV relative to CA.

### 4.2. Comparative Analysis of Hydrochemical Characteristics and Types

#### 4.2.1. Drinking Water

According to the requirements of the Standards for Drinking Water Quality (GB 5749-2006), Se, Hg, Cr^6+^, Cu, Zn, Cd and Pb were not detected in the two control groups. The detected concentrations of TDS, TH, As (heavy metal), Mn, NO_3_-N and COD_Mn_ are shown as follows: acceptable limits (AL) > CA > LLV. The high concentrations of TDS and TH will increase the risk of cardiovascular and cerebrovascular diseases [40]; and the heavy metal elements in the water can be accumulated by ingesting food, and exceed the health limits, which would interfere with normal physiological functions and increase cancer risk [41,42]. Therefore, after long-term intake and accumulation, the adverse effects of TDS, TH and As in drinking water was shown as CA > LLV. The low concentration characteristics of heavy metal elements in LLV’s drinking water is similar to that of other longevity areas in South China [43]. The concentrations of SO_4_^2−^ are shown as follows: AL > LLV > CA, and the concentrations of Fe, Cl^−^ and F^−^ are shown as follows: LLV > AL > CA. In LLV, the concentration of Fe (0.86 mg/L), Cl^−^ (326.1 mg/L) and F^−^ (2.10 mg/L) were 1.72 times, 1.09 times and 1.75 times higher than AL, respectively. However, these concentration levels are not sufficient to be associated with significant adverse health effects. Studies have shown that when the concentrations of Fe and Cl^−^ in drinking water are above 2 mg/L and 250 mg/L, this only causes changes in the taste and color of drinking water but has no significant adverse health effects [44]. However, when the concentration of F^−^ in drinking water is above 3.0 mg/L, long-term intake can adversely affect bone structure (Table 4) [45]. Based on the ANOVA, the drinking water quality components in LLV were little different from CA (F [1, 3] = 0.488; *p*-value = 0.191 (>0.05)). And yet, the inorganic components in the drinking water of LLV and many other longevity areas have some characteristics in common. In addition, experts have done much work on the process of the content of inorganic components in water environment change actually affecting human health and longevity [46,47]. In LLV, the characteristics of low TDS and heavy metal elements (Cr^6+^, Cd, Hg, As and Pb) in drinking water have the same characteristics as those of Xiayi longevity county and Hainan longevity district [11,48]. The drinking water of LLV has the characteristics of high Fe and low Mn, which was similar to the characteristics of drinking water in Chengyang longevity district, Qingdao [49].

#### 4.2.2. Agricultural Water

Seven main inorganic indexes (pH, Cl^−^, Hg, Cd, As, Cr^6+^ and Pb) were analyzed statistically in agricultural water samples using the *Farmland Irrigation Water Quality Standard* (GB5084-2005). Hg, Cr^6+^ and Cd were not detected in the two control groups; the concentrations of As and Pb are shown as follows: AL > CA > LLV. In CA, the concentration of Cl^−^ ranged from 34.8 mg/L to 2971 mg/L, with the average concentration of 670.7 mg/L. The average concentration of Cl^−^ was 3.35 times and 1.92 times higher than LLV and AL, respectively, indicating that the agricultural water sources of CA were subject to domestic sewage. In LLV, the seven indexes all met the standard requirements (Table 5). Based on the ANOVA, the agricultural water quality components in LLV were little different from CA (F [1, 12] = 0.672; *p*-value = 0.342 (>0.05)).

#### 4.2.3. Hydrochemistry Types

According to the Shukalev classification diagram [22] of groundwater samples in LLV and CA (Figure 2), the sampling points of Na^+^ + Ca^2+^ and Cl^−^ + HCO_3_^−^ types were the main groundwater hydrochemistry types of LLV and accounted for 75.0% of all sampling points. The sampling points of Na^+^ + Mg^2+^, Na^+^, Cl^−^ + SO_4_^2−^ and Cl^−^ types were the main groundwater hydrochemistry types of CA, and accounted for 80% of all sampling points. In CA, as the sampling sites gradually transitioned from the middle-upper reaches to lower reaches of the Keriya River (H12→H18→H25→H23), and the groundwater hydrochemistry types were HCO_3_·Cl-Na·Mg → HCO_3_·Cl-Na → Cl·HCO_3_·SO_4_-Na → Cl·SO_4_-Na type, indicating that the groundwater environment in the lower reaches of the Keriya River is more seriously polluted by agricultural non-point source pollution and domestic sewage than the middle-upper reaches [21]. From the analysis of hydrochemistry types, it can be preliminarily judged that the overall mineralization of groundwater in CA was higher than LLV, and the CA groundwater environment may have higher health risks than LLV. To further assess the water quality level of the two control groups and to determine any effects on the residents’ health-longevity, a comparative assessment of water quality and health risk should be conducted.

### 4.3. Comparative Assessment of Water Environment Quality

According to the results of the comparative analysis of water quality in LLV and CA, this study selected ten over-standard inorganic components as the main evaluation factors and established the corresponding evaluation factor sets: $U_{Evaluation\;factors}$ = {TH, TDS, Na^+^, SO_4_^2−^, Cl^−^, Fe, Mn, F^−^, I^−^, As} (Table 6).

Water quality was assessed using the FCEM, and the results are listed in Table 7 and delineated in Figure 3.

Table 7 shows that the overall water environment category of LLV was moderate quality. Two sampling points (50.0% of all points) were good quality, and two sampling points (50.0% of all points) were moderate quality. The LLV water quality was generally suitable for various purposes, while the overall water environment category of CA was very poor. Two sampling points (13.3% of all points) were excellent quality, one sampling point (6.7% of all points) was good quality, three sampling points (20.0% of all points) were moderate quality, and nine sampling points (60.0% of all points) were very poor and cannot be used safely as a drinking water source. As shown in Figure 3, the water samples with very poor quality were mainly distributed in the northwestern side of CA (downstream of the Keriya River and near the Taklimakan Desert). As the local groundwater flows from south to north, groundwater quality deteriorates gradually, and the water quality in the CA groundwater discharge area reaches the very poor category. In CA, the terrain north of the G315 National Road is relatively flatter and has a large amount of farmland. Therefore, agricultural non-point source pollution is the main cause of local groundwater quality deterioration [50].

Furthermore, the northwestern side of CA is close to the Taklimakan Desert and evaporation is significant, which is the natural factor causing deterioration of the local groundwater quality [51,52]. Water quality has an extremely significant positive relationship with the longevity index of local residents [11]. Overall, the LLV groundwater quality was generally better than CA, therefore, the LLV water environment was more conducive to human health than CA.

### 4.4. Comparative Assessment of Health Risk

According to the above analysis, the groundwater in the two control groups are both polluted, and the type of contaminants in the CA is more than that of LLV. Main contaminants of LLV are TDS, Na^+^, Cl^−^, Fe and F^−^, and the main contaminants of CA are TDS, Na^+^, Cl^−^, Fe, F^−^, TH, SO_4_^2−^, Mn, I^−^ and As. After comparison of the over-standard inorganic components in the samples of the two control groups the main hazardous effects of inorganic pollutants including physicochemical properties, toxicological toxic effects, etc. can be identified [53]. The inorganic indexes selected for human health risk assessment include non-carcinogens (including Cl^−^, Fe, F^−^, Mn and As) and carcinogen (As). The IARC and WHO stipulate that carcinogens also have non-carcinogenic risks [54]. The specific values of *RfD_nj_* and *q_ci_* are shown in Table 8 [55,56,57]. Combined with the pollution levels of the two control groups, this study selected the maximum acceptability risk level of human health recommended by USEPA as reference criteria (1.0 × 10^−4^ year^−1^).

Assessment results (Table 9) reveal that the risks of non-carcinogens in LLV and CA are mainly caused by Cl^−^, and the mean values are 2.22 × 10^−6^ and 5.08 × 10^−6^, respectively. The health risk ratio (mean _CA_/mean _LLV_ of Cl^−^) is 2.29, indicating that the non-carcinogenic risk (*HQ_CA-Cl_*^−^) caused by Cl^−^ in the drinking water of CA is 2.29 times higher than LLV. In terms of *HQ_n_*, the order of other indexes of LLV is F^−^ > Fe > Mn > As; the order of other indexes in CA is F^−^ > Fe > As > Mn. The *HQ_CA-F^−^,Fe,Mn_* caused by F^−^, Fe and Mn in the drinking water of CA are 1.26, 16.84 and 2.88 times higher than LLV, respectively. As was not detected in sampling of LLV, indicating the *HQ_LLV-As_* and carcinogen risk (*Risk_LLV-As_*) caused by As are relatively negligible in the LLV water environment. However, As is the main carcinogenic factor in the CA water environment, which ranges from nondetect to 9.63 × 10^−6^, with a mean value of 2.69 × 10^−6^. The highest *Risk_CA-As_* for a single water sample is H25 (9.63 × 10^−6^ per year), and the second highest point is H20 (6.13 × 10^−6^ per year). The sampling location of H25 is downstream of the Keriya River, surrounding which there are agricultural non-point source pollution and domestic pollution. Measures should be taken to reduce this risk. Similarly, sampling location H25 is located in the southern edge of the Taklimakan Desert, downstream of the Keriya River borderland. In many areas of China, water environment arsenic pollution is due to pesticide over-application, which is associated with higher risk of carcinogenic diseases [58].

The mean values of *HQ_n_* and *Risk_c_* are much higher for the over-standard inorganic components in the drinking water of CA than that of LLV. Similarly, the Total Health Risks (*HR_all_*) of over-standard inorganic pollutants in the water environment of LLV and CA ranged from 1.34 × 10^−6^ to 3.15 × 10^−6^ and 1.29 × 10^−6^ to 2.9 × 10^−5^, respectively. The *HR_all_* of all water samples in LLV and CA are lower than the acceptable level recommended by USEPA (this samples have potential acceptable health risks). However, the mean value of *HR_CA-all_* in CA is 3.49 times higher than LLV, indicating the health risk associated with over-standard inorganic pollutants in the CA drinking water is 3.49 times higher than that of LLV.

The health risk ratio (Figure 4) results of the two control groups show that the *HQ_LLV-Cl_^−^* caused by Cl^−^ accounts for more than 99.0% of the *HR_all_* in LLV. Among the water samples in CA, 53.3% of the *HR_all_* are mainly *Risk_CA-As_* caused by As (with a health risk ratio of As > 50.0%), and 46.7% of the *HR_all_* are mainly *HQ_CA-Cl_^−^* caused by Cl^−^. Moreover, the main non-carcinogen of the two control groups was Cl^−^, indicating the water environment of the two control groups had similar pollution sources. The domestic pollution discharged directly into the river channel by local residents where the Cl^−^ comes from in the groundwater in the study area. The Cl^−^ does penetrate into the groundwater aquifer through seepage [59]. In the downstream plain area of the Keriya River (CA), the *HR_all_* of H14, H15, H17, H18 and H25 are mainly caused by As, which is mainly due to the poor groundwater dynamic runoff conditions in the downstream [60]. In addition, the enrichment of As in groundwater is closely related to the use of arsenic-containing pesticides and fertilizers in agricultural activities [61].

Li et al. found that As in local groundwater was mainly affected by soil parent material, agricultural activities and household pollution in Hotan River Basin. The enrichment of As in groundwater is closely related to the use of excess amount of arsenic-containing pesticides and fertilizers in agricultural activities [62]. In general, residents living in CA are at higher health risk, especially in the downstream plain area of the Keriya River. Chen et al. recently found that As gives too high value in the surface soil of Yutian County where located in the middle and lower reaches of the Keriya River. The results were consistent with the spatial distribution characteristics of groundwater with higher arsenic concentration region in our research [63].

It should be noted that the health risk assessment results are to a certain degree uncertain. Since some residents of the two control groups often drink boiled groundwater, the concentration of some chemical components in the water also changes due to instability under high temperature conditions. In addition, this assessment did not consider other exposure pathways such as human body skin contact and inhalation.

## 5. Conclusions

The longevity area and non-longevity area of Xinjiang have the same climatic and hydrogeological conditions. Also, the residents of the two control groups have similar ethnic composition, diet and lifestyles. However, there are differences in the population life-span of the two control groups, in particular the longevity level parameters of the longevity area are higher than that of the non-longevity area. The difference between the water quality and human health risk in the two control groups is one of the factors influencing the life-span and health of residents. In the two control groups, the concentrations of TDS, TH and heavy metal elements (As) in drinking water and heavy metal elements (As and Pb) and Cl^−^ in agricultural water are CA > LLV. In addition the CA agricultural water source is affected by domestic sewage. The overall water environment categories in LLV and CA are moderate quality and very poor quality, respectively. In addition, the hydrochemistry types of LLV are mainly the Na^+^ + Ca^2+^ and Cl^−^ + HCO_3_^−^ types and the hydrochemistry types of CA are mainly the Na^+^ + Mg^2+^, Na^+^, Cl^−^ + SO_4_^2^^−^ and Cl^−^ types.

Compared to non-carcinogenic risk (*HQ_LLV_*), the carcinogenic risks (*Risk_LLV_*) is relatively negligible in LLV (non-carcinogen is mainly Cl^−^, carcinogen is not detected). The total health risks (*HR_LLV_*) of all water samples in LLV are lower than the acceptable level recommended by USEPA. The *HR_CA_* of the CA consisted of *HQ_CA_* (mainly caused by Cl^−^) and *Risk_CA_* (mainly caused by As), of which 46.7% of the sampling points showed *HQ_CA_* > *Risk_CA_*; 53.3% of the sampling points showed *Risk_CA_* > *HQ_CA_*. The *HR_CA_* of all CA water samples is lower than the acceptable level recommended by USEPA. However, the *HR_all_* caused by over-standard inorganic pollutants in CA is 3.49 times higher than that of LLV. Compared with LLV, residents in CA are not only exposed to the acceptable carcinogenic risk caused by As, but also exposed to higher non-carcinogenic risk, especially in the downstream plain area of the Keriya River. In the present study, inorganic contaminants with health risks are mainly derived from agricultural activities and domestic sewage. The LLV water quality is generally better than that of CA, and the total health risk in the LLV water environment is much lower than that of CA. Therefore, the LLV water environment is more conducive to the health and longevity of residents than CA.

We have linked water quality to human longevity in terms of cancer risk. This study was the discovery and preliminary investigations. We would like to study the possible link between local water environment and human longevity and healthy with endemic diseases and other common local diseases for future study. The longevity area and non-longevity area selected in this study have specific natural geographical conditions and arid climate conditions. Our research perspective might provide a scientific basis for the study of the relationship between water environment and health risks in arid-climate longevity areas.

## Figures and Tables

**Figure 1 ijerph-16-03737-f001:**
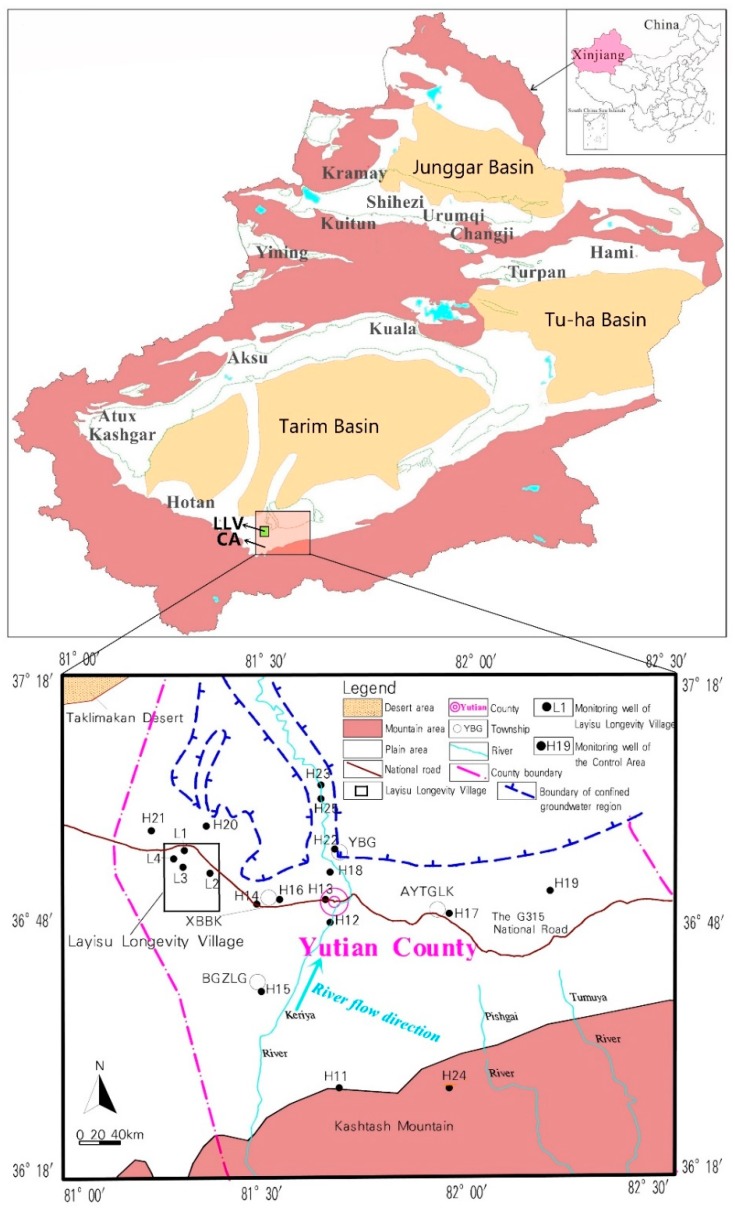
The distribution of sampling points of LLV and CA.

**Figure 2 ijerph-16-03737-f002:**
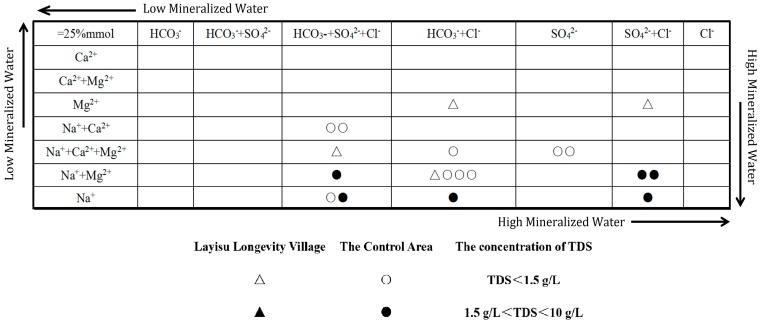
Shukalev classification diagram [21] of groundwater samples in LLV and CA.

**Figure 3 ijerph-16-03737-f003:**
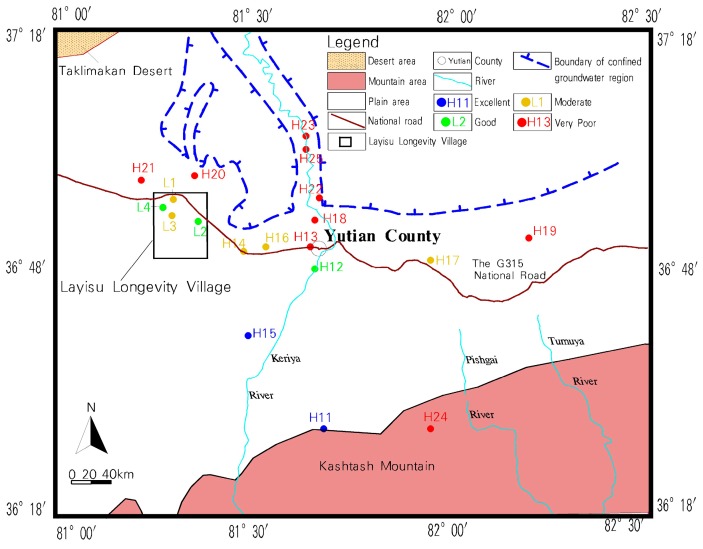
Results of water environment quality assessment on groundwater.

**Figure 4 ijerph-16-03737-f004:**
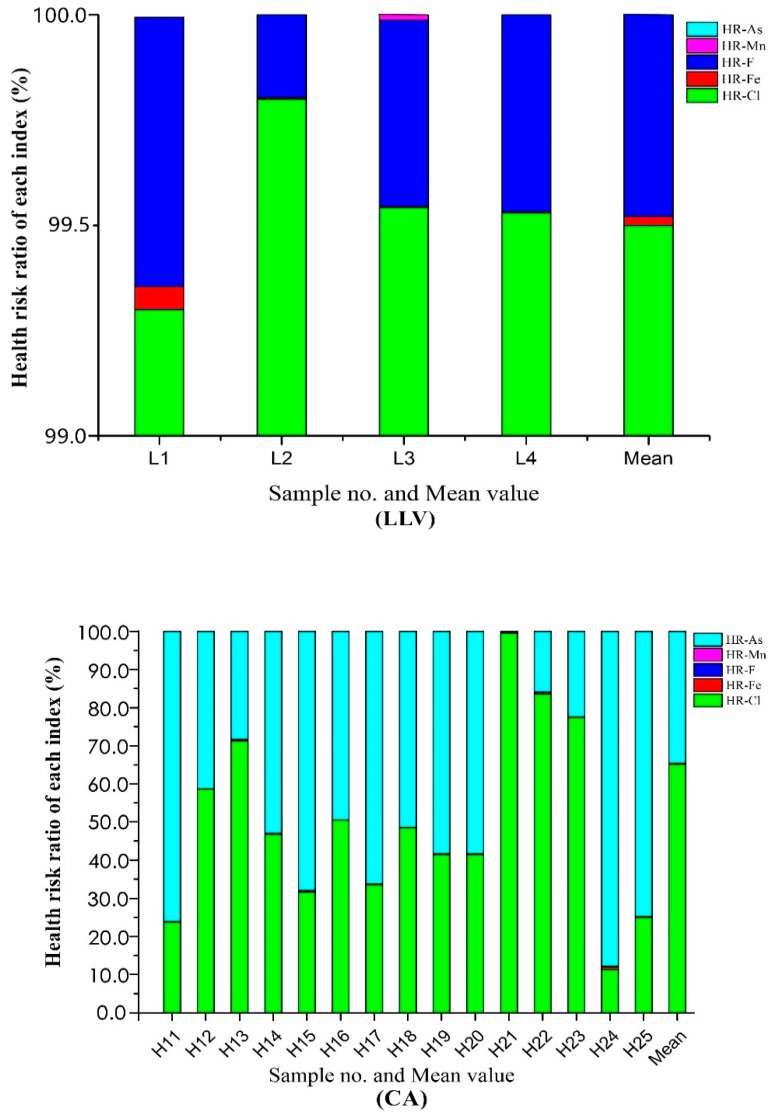
Health risk ratio of each index in the two control groups.

**Table 1 ijerph-16-03737-t001:** Comparison of the climate factors, hydrogeology conditions and economic contrasts.

Factors	LLV	CA
Climate condition	temperate continental arid climate	temperate continental arid climate
Annual average temperature	11.7 °C	11.6 °C
Annual precipitation	45.7 mm	47.7 mm
Annual evaporation	2460.3 mm	2432.1 mm
Atmospheric relative humidity	39.8%	42.0%
Average elevation	1350 m	1531 m
Frostless period	205~210 days	207~213 days
Groundwater buried condition	Unconfined groundwater	Unconfined groundwater
Per capita GDP	4085 yuan ($571)	5250 yuan ($733)
Per capita grain planting area	0.14 hm^2^	0.12 hm^2^

LLV: Layisu Longevity Village; CA: the adjacent control area.

**Table 2 ijerph-16-03737-t002:** Analytical measurement instruments, methods and detection limit of indexes.

Indexes	Measurement Instruments and Methods	Detection Limit (mg/L, Except pH Value)
pH	Portable digital pH meter MT-8060	0–14.000
TDS/TH	EDTA titration method	0.4/0.32
K^+^/Na^+^/Ca^2+^/Mg^2+^/free CO_2_	Ion chromatograph ICS1500	0.05/0.01/0.2/0.12/0.07
Li/Sr/Cl^−^/SO_4_^2−^/H_2_SiO_3_	Ion chromatograph ICS1500	0.003/0.00610/0.09/1
HCO_3_^−^	Titration method	5
I^−^/F^−^/TFe/Cu/Pb/Zn/Mn /Cr^6+^/Cd/Hg/As/Se/Br NH_4_^+^/NO_3_^−^/NO_2_^−^	Inductively coupled plasma atomic emission spectrometer (ICP-AES) iCAP 6300 Ultraviolet spectrophotometer UV2550	0.02/0.1/0.05/0.01/0.001/0.002/0.001 /0.004/0.002/0.0001/0.001/0.001/0.005 0.05/0.02/0.02

**Table 3 ijerph-16-03737-t003:** Comparative statistics of population status in LLV and CA.

Factors	LLV	CA
Population size	2922	28.65 × 10^4^
Agricultural population	2586	25.98 × 10^4^
Uygur population share	99.6%	98.3%
Population aged 60 and above	243	19769
Population aged 80 and above	24	1364
Population aged 100 and above	3	14
Average life expectancy * Centenarians/Total population	78.4 1.03 × 10^−3^	77.2 4.89 × 10^−5^
Child mortality rate **	3.05‰	12.01‰

Note: * from reference [36], ** from 2016 Department Decision Analysis Report of Yutian County Health Bureau.

**Table 4 ijerph-16-03737-t004:** Comparative statistical analysis of the mean values ± standard deviations of inorganic components in drinking water (mg/L, except pH value).

Indexes	pH	Cl^−^	F^−^	NO_3_-N	As	Se	Hg	SO_4_^2^^−^	Cr^6+^
Acceptable limits	≥6.5, ≤8.5	≤300	≤1.2	≤20	≤0.01	≤0.01	≤0.001	≤250	≤0.05
LLV (*N* = 1) CA (*N* = 4)	7.84 7.94 ± 0.3	326.1 111.5 ± 59.9	2.10 0.23 ± 0.2	0.52 4.73 ± 4.3	ND 0.002 ± 0.001	ND ND	ND ND	196.9. 162.4 ± 81.9	ND ND
Indexes	Fe	Mn	Cu	Zn	Cd	Pb	TDS	TH	COD_Mn_
Acceptable limits	≤0.5	≤0.3	≤1.0	≤1.0	≤0.005	≤0.01	≤1500	≤550	≤5.0
LLV (*N* = 1) CA (*N* = 4)	0.86 0.32 ± 0.4	ND 0.01 ± 0.01	ND ND	ND ND	ND ND	ND ND	651.6 1141.3 ± 279.6	326.7 430.3 ± 81.9	0.72 1.00 ± 0.1

Note: There is only one sampling site (sample no. L1) for drinking water in LLV, so there is no standard deviation; ND non detected.

**Table 5 ijerph-16-03737-t005:** Comparative statistical analysis of agricultural water quality indexes (mg/L, except pH value).

Indexes	Acceptable Limits	LLV (*N* = 3)			CA (*N* = 11)		
		Min	Max	Mean ± SD	Min	Max	Mean ± SD
pH	5.5~8.5	7.8	8.01	7.93 ± 0.1	7.11	8.7	7.87 ± 0.5
Cl^−^	≤350	140	269.4	200.3 ± 65.2	34.8	2971	670.7 ± 834.1
Hg	≤0.001	ND	ND	ND	ND	ND	ND
Cd	≤0.01	ND	ND	ND	ND	ND	ND
As	≤0.1	ND	ND	ND	ND	0.01	0.004 ± 0.002
Cr^6+^	≤0.1	ND	ND	ND	ND	ND	ND
Pb	≤0.2	ND	ND	ND	ND	0.006	0.002 ± 0.002

Note: Mean ± SD is mean values ± standard deviations; ND non detected.

**Table 6 ijerph-16-03737-t006:** Over-standard inorganic components and standard classification.

Index	Grade I	Grade II	Grade III	Grade IV	Grade V	Over-Standard Rate (%)	
						LLV (*N* = 4)	CA (*N* = 15)
Classification	Excellent	Good	Moderate	Poor	Very poor		
TH (mg/L)	≤150	≤300	≤450	≤650	>650	0	46.7
TDS (mg/L)	≤300	≤500	≤1000	≤2000	>2000	25	53.3
Na^+^ (mg/L)	≤100	≤150	≤200	≤400	>400	50	46.7
SO_4_^2^^−^ (mg/L)	≤50	≤150	≤250	≤350	>350	0	53.3
Cl^−^ (mg/L)	≤50	≤150	≤250	≤350	>350	50	53.3
Fe (mg/L)	≤0.1	≤0.2	≤0.3	≤2.0	>2.0	25	66.7
Mn (mg/L)	≤0.05	≤0.05	≤0.10	≤1.50	>1.50	0	46.7
F^−^ (mg/L)	≤1.0	≤1.0	≤1.0	≤2.0	>2.0	50	40
I^−^ (mg/L)	≤0.04	≤0.04	≤0.08	≤0.50	>0.50	0	13.3
As (mg/L)	≤0.001	≤0.001	≤0.01	≤0.05	>0.05	0	7

Note: Classification were derived from the Groundwater Quality Standard (GB/T14848-2017).

**Table 7 ijerph-16-03737-t007:** Comparative assessment result of water environment quality based on FCEM.

LLV (*N* = 4)							CA (*N* = 15)						
Sample No.	Grade I	Grade II	Grade III	Grade IV	Grade V	Assessment Results	Sample No.	Grade I	Grade II	Grade III	Grade IV	Grade V	Assessment Results
L1	0.059	0.075	0.495 *	0.223	0.193	Grade III	H11	0.426 *	0.205	0.226	0.220	0.000	Grade I
L2	0.342	0.528 *	0.130	0.000	0.000	Grade II	H12	0.164	0.707 *	0.129	0.000	0.000	Grade II
L3	0.021	0.172	0.733 *	0.074	0.000	Grade III	H13	0.015	0.004	0.048	0.001	0.933 *	Grade V
L4	0.218	0.431 *	0.352	0.000	0.000	Grade II	H14	0.015	0.128	0.495 *	0.337	0.026	Grade III
MC	0.015	0.402	0.564 *	0.020	0.000	Grade III	H15	0.557 *	0.021	0.236	0.187	0.000	Grade I
							H16	0.040	0.351	0.609 *	0.000	0.000	Grade III
							H17	0.228	0.281	0.456 *	0.036	0.000	Grade III
							H18	0.054	0.054	0.087	0.011	0.795 *	Grade V
							H19	0.163	0.099	0.210	0.192	0.338 *	Grade V
							H20	0.002	0.003	0.046	0.046	0.900 *	Grade V
							H21	0.005	0.000	0.010	0.002	0.804 *	Grade V
							H22	0.000	0.008	0.012	0.000	0.981 *	Grade V
							H23	0.006	0.004	0.106	0.085	0.800 *	Grade V
							H24	0.082	0.004	0.025	0.002	0.887 *	Grade V
							H25	0.006	0.044	0.266	0.138	0.537 *	Grade V
							MC	0.008	0.004	0.052	0.136	0.799 *	Grade V

FCEM: Fuzzy Comprehensive Evaluation Method, MC: Mean concentration of all sample points, Number *: Maximum weight of the five levels.

**Table 8 ijerph-16-03737-t008:** Reference dosage values of non-carcinogens and cancer slope factor of carcinogen.

Parameters	As	Cl^−^	Fe	F^−^	Mn
*RfD_nj_* [mg/(kg·day)]	3.0 × 10^−4^	0.06	0.3	0.06	0.14
*q_ci_* [(kg·day)/mg]	1.5				

**Table 9 ijerph-16-03737-t009:** Human health risk assessment result of LLV and CA (per year).

Control Groups and Samples No.		Non-Carcinogens					Carcinogen	*HQ_n_*	*Risk_c_*	*HR_all_*
		As	Cl^−^	Fe	F^−^	Mn	As			
LLV	L1	ND	3.13 × 10^−6^	1.70 × 10^−9^	2.01 × 10^−^^8^	ND	ND	3.15 × 10^−6^	ND	3.15 × 10^−6^
	L2	ND	1.34 × 10^−6^	ND	2.88 × 10^−9^	ND	ND	1.34 × 10^−6^	ND	1.34 × 10^−6^
	L3	ND	2.58 × 10^−6^	ND	1.15 × 10^−^^8^	3.29 × 10^−10^	ND	2.59 × 10^−6^	ND	2.59 × 10^−6^
	L4	ND	1.83 × 10^−6^	ND	8.63 × 10^−9^	ND	ND	1.84 × 10^−6^	ND	1.84 × 10^−6^
	Mean	ND	2.22 × 10^−6^	4.60 × 10^−10^	1.08 × 10^−^^8^	1.59 × 10^−10^	ND	2.23 × 10^−6^	ND	2.23 × 10^−6^
CA	H11	3.89 × 10^−9^	5.49 × 10^−7^	2.28 × 10^−9^	2.14 × 10^−9^	1.17 × 10^−10^	1.75 × 10^−6^	5.57 × 10^−7^	1.75 × 10^−6^	2.31 × 10^−6^
	H12	1.95 × 10^−9^	1.25 × 10^−6^	4.07 × 10^−10^	4.87 × 10^−^^10^	1.67 × 10^−11^	8.76 × 10^−7^	1.25 × 10^−6^	8.76 × 10^−7^	2.13 × 10^−6^
	H13	3.89 × 10^−9^	4.42 × 10^−6^	6.33 × 10^−10^	2.34 × 10^−^^8^	5.72 × 10^−10^	1.75 × 10^−6^	4.45 × 10^−6^	1.75 × 10^−6^	6.20 × 10^−6^
	H14	5.84 × 10^−9^	2.32 × 10^−6^	2.14 × 10^−11^	1.65 × 10^−^^8^	4.21 × 10^−10^	2.63 × 10^−6^	2.35 × 10^−6^	2.63 × 10^−6^	4.97 × 10^−6^
	H15	1.95 × 10^−9^	4.09 × 10^−7^	2.05 × 10^−9^	3.11 × 10^−9^	1.08 × 10^−10^	8.76 × 10^−7^	4.16 × 10^−7^	8.76 × 10^−7^	1.29 × 10^−6^
	H16	3.89 × 10^−9^	1.79 × 10^−6^	3.31 × 10^−11^	1.56 × 10^−9^	ND	1.75 × 10^−6^	1.80 × 10^−6^	1.75 × 10^−6^	3.55 × 10^−6^
	H17	3.89 × 10^−9^	8.91 × 10^−7^	2.92 × 10^−11^	3.89 × 10^−9^	ND	1.75 × 10^−6^	8.99 × 10^−7^	1.75 × 10^−6^	2.65 × 10^−6^
	H18	9.73 × 10^−9^	4.14 × 10^−6^	1.21 × 10^−9^	6.42 × 10^−9^	2.71 × 10^−10^	4.38 × 10^−6^	4.16 × 10^−6^	4.38 × 10^−6^	8.54 × 10^−6^
	H19	3.89 × 10^−9^	1.25 × 10^−6^	7.40 × 10^−11^	4.28 × 10^−9^	ND	1.75 × 10^−6^	1.26 × 10^−6^	1.75 × 10^−6^	3.01 × 10^−6^
	H20	1.36 × 10^−8^	8.30 × 10^−6^	2.67 × 10^−9^	5.26 × 10^−^^8^	8.72 × 10^−10^	6.13 × 10^−6^	8.37 × 10^−6^	6.13 × 10^−6^	1.45 × 10^−5^
	H21	ND	2.89 × 10^−5^	6.52 × 10^−8^	6.62 × 10^−9^	1.73 × 10^−9^	ND	2.90 × 10^−5^	ND	2.90 × 10^−5^
	H22	3.89 × 10^−9^	9.26 × 10^−6^	3.97 × 10^−9^	5.45 × 10^−^^8^	4.09 × 10^−10^	1.75 × 10^−6^	9.32 × 10^−6^	1.75 × 10^−6^	1.11 × 10^−5^
	H23	5.84 × 10^−9^	9.09 × 10^−6^	8.37 × 10^−^^10^	1.36 × 10^−^^8^	4.51 × 10^−10^	2.63 × 10^−6^	9.11 × 10^−6^	2.63 × 10^−6^	1.17 × 10^−5^
	H24	5.84 × 10^−9^	3.39 × 10^−7^	2.20 × 10^−^^8^	2.14 × 10^−9^	9.43 × 10^−10^	2.63 × 10^−6^	3.70 × 10^−7^	2.63 × 10^−6^	3.00 × 10^−6^
	H25	2.14 × 10^−8^	3.23 × 10^−6^	1.48 × 10^−^^8^	1.27 × 10^−^^8^	9.64 × 10^−10^	9.63 × 10^−6^	3.28 × 10^−6^	9.63 × 10^−6^	1.29 × 10^−5^
	Mean	5.97 × 10^−9^	5.08 × 10^−6^	7.75 × 10^−9^	1.36 × 10^−^^8^	4.58 × 10^−10^	2.69 × 10^−6^	5.11 × 10^−6^	2.69 × 10^−6^	7.79 × 10^−6^
Mean _CA_/Mean _LLV_		-	2.29	16.84	1.26	2.88	-	2.29	-	3.49

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
