# Peer review of "A Comparative Study of Water Quality and Human Health Risk Assessment in Longevity Area and Adjacent Non-Longevity Area"

_ijerph, 2019, doi:10.3390/ijerph16193737_

Round 1

Reviewer 1 Report

1. The manuscript “Comparative Study of Water Quality and Human Health Risk Assessment in Longevity Area and Adjacent Non-Longevity Area, Xinjiang, China” investigated if differences in groundwater quality between the longevity area and the non-longevity area are associated with the health of residents in the two control groups using the Fuzzy Comprehensive Evaluation Method, which is interesting. It is relevant and within the scope of the journal.

2. However the manuscript, in its present form, contains several weaknesses. Appropriate revisions to the following points should be undertaken in order to justify recommendation for publication.

3. For the formula 2, whether it's the sentence “Water quality from grade 2 to (m-1) is:” or the sentence “Water quality from grade 2 to (n-1) is:” needs to be checked.

4. Figures and tables should be standardized and clear, such as Figure 1, and Figure 2 should be a table, etc. needs to be modified.

5. For readers to quickly catch your contribution, it would be better to highlight major difficulties and challenges, and your original achievements to overcome them, in a clearer way in abstract and introduction.The discussion section in the present form is relatively weak and should be strengthened with more details and justifications.

6. Moreover, the fuzzy method used in this paper is an intelligent method. At present, there are many other intelligent evaluation methods in the field of water environment, such as set pair analysis method, optimization method, multi-objective evaluation method and so on. The manuscript could be substantially improved by relying and citing more on recent literatures about real-life case studies of contemporary intelligent techniques in elsewhere such as the followings:
Yang X.H., Zhang X.J., Hu X.X., Nonlinear Optimization Set Pair Analysis Model (NOSPAM) for Assessing Water Resource Renewability. Nonlinear Processes in Geophysics, 2011,18:599-607;
K.W. Wang, X.H. Yang, X.M. Liu, et al., A simple analytical infiltration model for short-duration rainfall, Journal of Hydrology, 555 (2017) 141-154;
Xiao-Hua Yang, Chong-Li Di, Jun He, Jian Zhang and Yu-Qi Li, Integrated assessment of water resources vulnerability under climate change in Haihe River Basin, International Journal of Numerical Methods for Heat and Fluid Flow,2015,25(8): 1834-1844

The paper is acceptable for publication with minor revisions.

Author Response

We very much appreciate your valuable suggestions and professional revision of our manuscript (Manuscript ID ijerph-580132). We have carefully considered the comments and have revised the manuscript accordingly. Please see the attachment.

Reviewer 2 Report

This work conducted a comparative study of the longevity area and the adjacent control area, aiming to find the factors that contribute to the longevity of the local population. The experiments are well designed and the results reasonably interpreted, and the conclusion is generally convincing. The authors should pay attention to the following issues:

The authors should indicate more information like the flow direction of the rivers and groundwater, farmland or industrial area, et al. on the sampling map for better understanding of the results. The selection and distribution of the sampling points should be explained more. e.g., why most of the points are along with the national road? The parameters such as the longevity level index, the 100-year-old level index should be defined in the text (line 178-184). line 261: Table 9? should be a mistake? line 327-333: about the pollution source of Cl- and As, the authors should offer more experimental or statistical information of the local environment to indicate more clearly the sources of pollutants. And what is the relation between the Keriya River and groundwater pollution, as well as the local agricultural activities and domestic sewage discharge? Now these issues are not explained clearly in the text. Fig. 4: please mind that the legends and the tick labels are incorrect.

Author Response

Thank you very much for your careful reading and excellent comments of our manuscript (Manuscript ID ijerph-580132). We have carefully considered the comments and have modified the manuscript accordingly. Please see the attachment.

Reviewer 3 Report

The manuscript discussed a very interesting topic,the relationship between groundwater quality and human life. The well-known Layisu Longevity Village and an adjacent control area were selected, investigated and comparatively assessed. And as a result,higher water quality and lower healthy risks of groundwater were tested in the longevity area compared with the control aera.

Several improments will be necessary before publication:

The selection of the control area was not reasonable enough. The control area covers nearly 100 times larger than the target village, with different population distribution and quite varibale water quality, which was clearly supported in table 8 and figure 4. Evidents were not found in this manuscripyt that high groundwater quality led to longevity. The significance of this work will be deeply weakened if the author fails to establish the connectin between the water quality and the human longevity. Several small leaks in the figures, such as the river flow direction in figure 1 and the legends of the control data in figure 4.

Author Response

Thank you very much for your attention and helpful comments of our manuscript (Manuscript ID ijerph-580132). We have carefully considered the comments and have modified the manuscript accordingly. Please see the attachment.

Reviewer 4 Report

The hypothesis is exciting and potentially important. However, the authors did not analyze satisfactorily to test the hypothesis. There are a lot of issues that could be responsible for the differences in longevity in two adjacent places – authors need to touch these issues as a backdrop to make a case for their hypothesis. The methodology and data collections section also need substantial work to help the readers to understand the analysis better.

I am wondering whether the hypothesis used in this study clearly justifies the issue. From Table 1, it is apparent that the climatologically and hydrogeologically there are not much difference between the sites. I would like to see the various economic contrasts between the two control areas in this table. It is also tough to understand what population data was used and how they were obtained. Authors cited Mattimin and Eli, 2015 to describe the data. However, the population data defined in line 177 to 187 can be represented by a table for better understanding. Authors also need to mention the year of the collected data.  Another concern is whether the groundwater (GW) data were collected concurrently with the population data. Since the GW data was collected in 2014, and 2017, the population data should also represent a similar or closely similar time window.

I would like to see more statistical analysis with the data – use of ANOVA to statistically determine whether the water quality parameters differ between the sites.

I prefer to see more in-depth discussion rather than the texts describing the results that are reported in the form of tables and figures. The current version of the manuscript manly provides the results. The obtained results and hypothesis need to be compared with the existing literature – whether similar observations were made that linked age expectancy with the water environment.

The title may also need revision; the current title does not necessarily reflect the objectives and findings of the study.

line 58: experiences.

line 76: add a space before m3/a. Does a indicate acre? Please spell it out.

Table 1: What is the source of data in Table 1? 

Figure 1: Figure 1 is difficult to follow. What is the boundary of the CA?

Are the 15 groundwater sample points located within the CA? or those were within the watershed of the CA area? Are LLV and CA share the same watershed? The authors need to describe thy watershed processes better here.

How many samples are collected in each collection point? What is the data collection uncertainty? It looks like the GW data for the LLV and CA were not collected at the same time-- can this contribute to the results?

Please comment whether any QA/QC was performed with the measured data? Also, it would be great to see a box plot/violin plot of the information for a better understanding.

Please indicate the software package (R/Matlab?) that was used to apply the fuzzy method? 

Please define longevity index.

Eq. 4 and 5: In the method section, it is essential to describe the data used to calculate the risks?

Section 4.1: I don’t see much difference in average life expectancy between the two controls? What is the child mortality rate of the areas which can contribute to the average life expectancy?

Refer Table 1 in line 209 where it was first mentioned. Are those average values across sampling points? If yes, please indicate that in the table along with standard errors or standard deviations of the averages.

Table 3 caption: there is no statistical analysis here rather than the reported average values. Please write the table caption. Also, please indicate the sample size.

What is the difference between agricultural and drinking samples? How they differ? These appeared suddenly in the result. Please describe those sampling points in the method sections, whether they are agricultural or drinking point source.

Please describe the sources that were used to classify inorganic components in Table 5.

The assessment results reported in Table 6 are shown again in Figure 3. This is redundant, and either of them could be relocated in Supplemental note. What is the units (mg/l?) of the grades reported in Table 6. Also, please report the uncertainty of MC in the form of standard error or standard deviation.

Table 4: Indicate the sample size and standard error of the data? Which data points (H11-25/L1-L4) they represent?

Where is Table 9, as indicated in line 261?

Line 324: How the total health risk ration is calculated?

Figure 4 needs much revision. In the y-axis of the first plot, 100% appears twice. Further, the axis captions overlapped with the X-axis values. The legends used to represent HR-AS, HR-Mn, and HR-F were not current. There is no need to provide the % sign for all the numbers. Simply, the sign can be used as a parenthesis in the y-axis caption.  

Line 342: Conclusions.

Author Response

(The authors gave the same response as above.)

Round 2

Reviewer 3 Report

Significant progress has been made in the revised manuscript. Although "Response 2" was not very convincing, the manuscript provided a meaningful research work. Overall, I agree to publish the manuscript .

Reviewer 4 Report

The revised version seems much better. However, I still have a concern that I have raised in my previous review. To my understanding, there are not many differences in terms of longevity between LLV and CA. The average life expectancy of LV was 1.2 years higher than the CA. The mean population age difference is also less than two years. Given the higher infant mortality rates of CA and the comparatively large population size (Table 2), this difference is not substantial.

Further, the reported life expectancy of LV and CA are higher than the global average, and most importantly, the values are higher than the average life expectancy of China. Further, percent of the population over 80 years between the two control groups are also comparable (0.47 vs. 0.81%). Therefore, based on the data, definition of CA as a non-longevity area is questionable, which makes the entire hypothesis that links water quality to longevity based on the data from two control groups weak. I agree that there is a difference in water quality between the sites, but the hypothesis gets thin because of the intangible differences in longevity. Can the authors present any literature evidence that defines the CA as a non-longevity area?

I did not see any results of ANOVA of the drinking water sampling point in the revised manuscript, although the authors explained the ANOVA results in response to the reviewer document.

Lines 224-225: What is “perfectly good research”- please revise the sentence.
Line 62: Should be [14-16].

About the QA/QC, I understand the data collection met all standard criteria. In line 232, authors indicated that there only one sampling site (L1) from where drinking water quality data were collected. I would appreciate if the authors describe the purpose of L2-L4 sites in LLV. I guess L2-L4 are for agricultural water. Further, are those samples just collected for once, or multiple samples were collected for a particular location? In general, just one random sample from a single site indicates a very high uncertainty.

I do not think a full stop is needed after ND (non detected).

Table 5: Title should be agricultural water quality indexes. Why please revise Agricultural to agricultural.

Figure 2: Where is the source of Figure 2? Do the authors prepare the figure? If not, please indicate the source.

Table 6: Standard should be standard.

Table 7: Assessment should be assessment.

Round 3

Reviewer 4 Report

First of all, line numbers that authors indicated in response to reviewers document did not match with the manuscript. Secondly, I do not think the differences between the CA and LLV are substantial, as the authors claimed. I think the differences are relative – authors also acknowledged that in response to review document. I strongly suggest the authors revise the manuscript to reflect the relative differences in longevity between the sites. The good thing is that it is already written in response to reviewer document as follows:

However, the relative population aggregation of centenarians in LLV is much higher than that in CA. In this paper, the longevity area with the non-longevity area is the concept of a "relative". This can be seen to result from the comparative analysis of the research results by Wang et al., (2011) "Research progress on factors influencing longevity", Maimaiti et al., (2015) "Current situation and characteristics of population development in Hotan area" and Xiong et al., (2007) "Population change and sustainable development in Yutian County, Xinjiang".

Please add some summarized texts from the above response note to the main text to build your argument. 

The texts associated with ANOVA was not good. Please review the literature to see how it is written. For example, it can be something like this: Based on the ANOVA, the drinking water quality components in LLV were significantly different from CA (F [DF, DF] = XX; P-value>0.05). ANOVA cannot show whether the differences are little or substantial as the authors indicated in the manuscript. It only indicates whether there is a significant difference in means between the two processes. Authors can also see the below link as a reference to write the ANOVA results: 

http://statistics-help-for-students.com/How_do_I_report_a_1_way_between_subjects_ANOVA_in_APA_style.htm#.XYjZKUZKguU

Also, the revision of the line 249-250 [And yet, experts have done much work on the process of water quality change actually affecting human health and longevity] does not read well. The line does not connect with the previous sentences. 
